# Simplifying Hamiltonian and Lagrangian Neural Networks via Explicit Constraints

**Marc Finzi**[*]
New York University

**Ke Alexander Wang**[*]
Cornell University

**Andrew Gordon Wilson**
New York University

## Abstract

Reasoning about the physical world requires models that are endowed with the right inductive biases to learn the underlying dynamics. Recent works improve generalization for predicting trajectories by learning the Hamiltonian or Lagrangian of a system rather than the differential equations directly. While these methods encode the constraints of the systems using generalized coordinates, we show that embedding the system into Cartesian coordinates and enforcing the constraints explicitly with Lagrange multipliers dramatically simplifies the learning problem. We introduce a series of challenging chaotic and extended-body systems, including systems with $N$-pendulums, spring coupling, magnetic fields, rigid rotors, and gyroscopes, to push the limits of current approaches. Our experiments show that Cartesian coordinates with explicit constraints lead to a 100x improvement in accuracy and data efficiency.

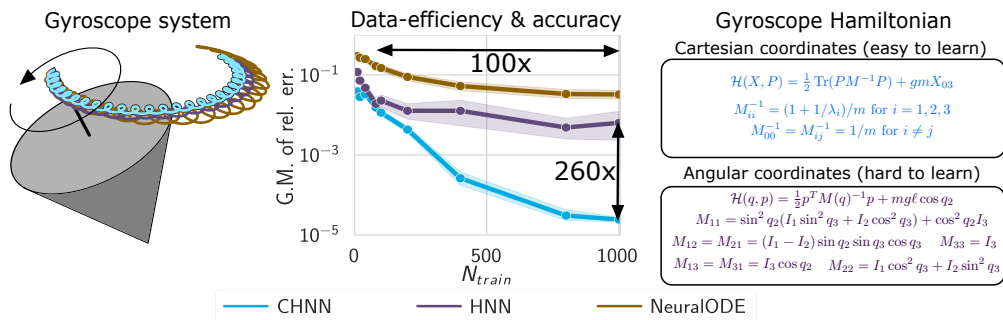

Figure 1: By using Cartesian coordinates with explicit constraints, we simplify the Hamiltonians and Lagrangians that our models learn, resulting in better long term predictions and data-efficiency than Neural ODEs and Hamiltonian Neural Networks (HNNs). **Left:** a spinning gyroscope with the ground truth trajectory and predictions of each model. Predicted trajectories by our model (CHNN) overlaps almost exactly with the ground truth (black). **Middle:** Geometric mean of the relative error over 100 timesteps as a function of number of training trajectories. On the gyroscope system, our model can be 100 times more data efficient or 260 times more accurate. **Right:** The Hamiltonian expressed in Cartesian coordinates is simpler and easier to learn than when expressed in angular coordinates.

## 1 Introduction

Although the behavior of physical systems can be complex, they can be derived from more abstract functions that succinctly summarize the underlying physics. For example, the trajectory of a physical system can be found by solving the system's differential equation for the state as a function of time.

---

[*]Equal contribution.

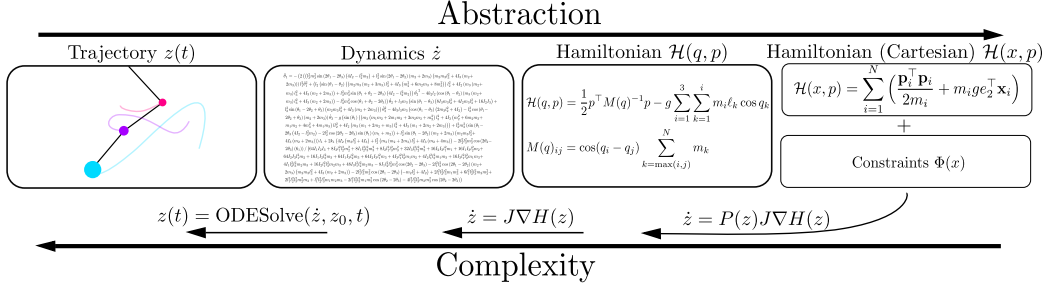

Figure 2: A visualization of how abstracting the physical system reduces the complexity that our model must learn. For systems like the 3-Pendulum, the trajectory is so complex that there is no closed form solution. Although the dynamics $\dot{z}$ do have a closed form, they require a long description. The Hamiltonian $\mathcal{H}$ of the system is simpler, and modeling at this higher level of abstraction reduces the burden of learning. Separating out the constraints from the learning problem, the Hamiltonian for CHNNs is even more succinct.

For many systems, these differential equations can in turn be derived from even more fundamental functions known as Hamiltonians and Lagrangians. We visualize this hierarchy of abstraction in Figure 2. Recent work has shown that we can model physical systems by learning their Hamiltonians and Lagrangians from data [9, 14, 20]. However, these models still struggle to learn the behavior of sophisticated constrained systems [2, 3, 7–9, 22]. This raises the question of whether it is possible to improve model performance by further abstracting out the complexity to make learning easier.

Constraints in physical systems are typically enforced by *generalized coordinates*, which are coordinates formed from any set of variables that describe the complete state of the system. For example, the 2D 2-pendulum in Figure 3 can be described by two angles relative to the vertical axis, labelled as $q = (q_1, q_2)$, instead of the Cartesian coordinates $x$ of the masses. By expressing functions in generalized coordinates, we ensure that constraints, like the distances from each pendulum mass to its pivot, are always satisfied implicitly. However, if we have a mechanism to explicitly enforce constraints, we can instead use Cartesian coordinates, which more naturally describe our three dimensional world.

In this paper, we show that generalized coordinates make the Hamiltonian and the Lagrangian of a system difficult to learn. Instead, we propose to separate the dual purposes played by generalized coordinates into independent entities: a state represented entirely in Cartesian coordinates $x$, and a set of constraints $\Phi(x)$ that are enforced explicitly via Lagrange multipliers $\lambda$. Our approach simplifies the functional form of the Hamiltonian and Lagrangian and allows us to learn complicated behavior more accurately, as shown in Figure 1.

In particular, our paper makes the following contributions. **(1)** We demonstrate analytically that embedding problems in Cartesian coordinates simplifies the Hamiltonian and the Lagrangian that must be learned, resulting in systems that can be accurately modelled by neural networks with 100 times less data. **(2)** We show how to learn Hamiltonians and Lagrangians in Cartesian coordinates via explicit constraints using networks that we term Constrained Hamiltonian Neural Networks (CHNNs) and Constrained Lagrangian Neural Networks (CLNNs). **(3)** We show how to apply our method to arbitrary rigid extended-body systems by showing how such systems can be embedded purely into Cartesian coordinates. **(4)** We introduce a series of complex phys-

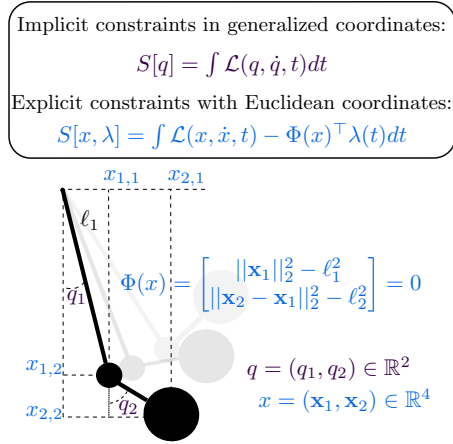

Figure 3: A 2D 2-pendulum expressed in terms of generalized coordinates $q$ and Cartesian coordinates $x$ with explicit constraints $\Phi(x) = 0$ for the Lagrangian formalism and the constrained Lagrangian formalism. $\mathcal{L}$ is the Lagrangian, a scalar function that summarizes the entire behavior of the system, entries of $\lambda$ are the Lagrange multipliers, and $S$ is a functional that is minimized by the system's true trajectory.

ical systems, including chaotic and 3D extended-body systems, that challenge current approaches to learning Hamiltonians and Lagrangians. On these systems, our explicitly-constrained CHNNs and CLNNs are 10 to 100 times more accurate than HNNs [9] and DeLaNs [14], which are implicitly-constrained models, and more data-efficient. Code for our experiments can be found at: https://github.com/mfinzi/constrained-hamiltonian-neural-networks.

## 2 Background on learning dynamical systems

An ordinary differential equation (ODE) is a system of differential equations which can be described by $\dot{z} = f(z, t)$ where $z(t) \in \mathbb{R}^D$ is the state as a function of time $t$ and $\dot{z}$ is shorthand for $dz/dt$. $f$ is known as the *dynamics* of the system since it alone specifies how the state changes with respect to time. A neural network $f_\theta$ can approximate the dynamics $f$ by learning from trajectory data $z(t)$ [1]. We can make predictions $\hat{z}(t)$ by integrating, $\hat{z}(t) = \text{ODESolve}(z_0, f_\theta, t)$, and compute the gradients of the loss function $L(\theta; z, \hat{z})$ with ordinary backpropagation or the adjoint method [1].

For many physical systems, the differential equations can be derived from one of two scalar functions, a Hamiltonian $\mathcal{H}$ or a Lagrangian $\mathcal{L}$, depending on the formalism. For example, the differential equations of a Hamiltonian system can be written as

$$\dot{z} = J\nabla\mathcal{H}(z), \qquad \text{where} \qquad J = \begin{bmatrix} 0 & I_{D/2} \\ -I_{D/2} & 0 \end{bmatrix}. \tag{1}$$

In this context the state $z = (q, p)$ is a concatenation of the *generalized coordinates* $q \in \mathbb{R}^{D/2}$ and the *generalized momenta* $p \in \mathbb{R}^{D/2}$ which parametrize the system's states on a manifold. The differential equations of a Lagrangian system can be written in a similar way except that they are only expressed in terms of $q$ and $\dot{q}$. Typically, $\dot{q}$ and $p$ are related by $p = M(q)\dot{q}$, a generalization of momentum from introductory physics, $p = m\dot{x}$, where $M(q)$ is the *mass matrix*.

Recent approaches predict trajectories by learning $\mathcal{H}$ or $\mathcal{L}$ instead of $f$. Greydanus et al. [9] proposed Hamiltonian Neural Networks (HNNs) which parametrize $\mathcal{H}$ with a neural network. Concurrently, Lutter et al. [14] proposed to learn $\mathcal{L}$ with Deep Lagrangian Networks (DeLaNs), which was used in robotics applications with additional control inputs. There are two main advantages of this approach: (1) the network only has to learn a scalar function, $\mathcal{L}$ or $\mathcal{H}$, whose functional form is simpler than the dynamics $f$, and (2) integrating the differential equations derived from the learned approximations $\mathcal{L}_\theta$ or $\mathcal{H}_\theta$ will result in trajectory predictions that better conserve energy since the true dynamics governed by $\mathcal{L}$ and $\mathcal{H}$ conserve energy. Naively, learning $\mathcal{H}$ requires training trajectories with states $z = (q, p)$. However, in real systems, the states are more commonly available in the form $z = (q, \dot{q})$. This inconvenience can be addressed for most systems since $p = M(q)\dot{q}$ and we can learn $M$ to convert between $\dot{q}$ and $p$ [20].

## 3 Related work

In addition to the work on learning physical systems above, Chen et al. [2] showed how symplectic integration and recurrent networks stabilize Hamiltonian learning including on stiff dynamics. Finzi et al. [7] showed how learned dynamics can be made to conserve invariants such as linear and angular momentum by imposing symmetries on the learned Hamiltonian. Zhong et al. [21] showed how to extend HNNs to dissapative systems, and Cranmer et al. [3] with LNNs showed how DeLaNs could be generalized outside of mechanical systems such as those in special relativity.

Our method relies on explicit constraints to learn Hamiltonians and Lagrangians in Cartesian coordinates. Constrained Hamiltonian mechanics was developed by Dirac [5] for canonical quantization — see Date [4] for an introduction. The framework for constrained Lagrangians is often used in physics engines and robotics [6, 17] — see LaValle [10] for an introduction. However, our paper is the first to propose learning Hamiltonians and Lagrangians with explicit constraints. Our approach leads to two orders of magnitude improvement in accuracy and sample efficiency over the state-of-the-art alternatives, especially on chaotic systems and 3D extended-body systems.

## 4 Simplifying function approximation with a change of coordinates

Previous works express the position of a system using generalized coordinates $q$, which has the advantage of automatically satisfying constraints, as explained in Section 1. However, the convenience of using generalized coordinates comes at the cost of making $\mathcal{H}$ and $\mathcal{L}$ harder to learn. These complications disappear when we embed the system in the underlying Cartesian coordinates.

We use a simple example to demonstrate how Cartesian coordinates can vastly simplify the functions that our models must learn. Suppose we have a chain of $N$ pendulums $i = 1, ..., N$ with point masses $m_i$ in 2D subject to a downward gravitational acceleration $g$. Indexing from top to bottom, pendulum $i$ is connected to pendulum $i - 1$ by a rigid rod of length $\ell_i$, as shown in Figure 3.

In Cartesian coordinates, the Hamiltonian and Lagrangian are simply

$$\mathcal{H}(x, p) = \sum_{i=1}^{N} \left[ \frac{\mathbf{p}_i^\top \mathbf{p}_i}{2m_i} + m_i g e_2^\top \mathbf{x}_i \right] \quad \text{and} \quad \mathcal{L}(x, \dot{x}) = \sum_{i=1}^{N} \left[ \frac{m_i}{2} \dot{\mathbf{x}}_i^\top \dot{\mathbf{x}}_i - m_i g e_2^\top \mathbf{x}_i \right],$$

where we used bold to denote the spatial vectors for the position and momentum $\mathbf{x}_i \in \mathbb{R}^2$ and $\mathbf{p}_i \in \mathbb{R}^2$ of mass $i$ respectively. Here $x$ and $p$ are concatenations of $\mathbf{x}_i$ and $\mathbf{p}_i$ over $i$.

We can also describe the system with generalized coordinates which implicitly encode the constraints. In this case, let $q_i$ be the angle of pendulum $i$ relative to the negative $y$ axis and $p_i$ the corresponding generalized momentum. In these coordinates, the Hamiltonian is

$$\mathcal{H}(q, p) = \frac{1}{2} p^\top M(q)^{-1} p - g \sum_{i=1}^{N} \sum_{k=1}^{i} m_i \ell_k \cos q_k \tag{2}$$

where the mass matrix has a complicated form with entries $M(q)_{ij} = \cos(q_i - q_j) \ell_i \ell_j \sum_{k=\max(i,j)}^{N} m_k$ which we derive in Appendix F.2. The Lagrangian is the same as Equation 2 except that the first term is replaced by $\dot{q}^\top M(q) \dot{q}/2$ and the second term is negated.

The expression in Cartesian coordinates is *linear* in the state $x$ and quadratic in $p$ and $\dot{x}$ with a *constant* and *diagonal* mass matrix with entries $M_{ii} = m_i$, while the expression in angular coordinates is nonlinear in $q$ and has off diagonal terms in $M(q)$ that vary with time as $q$ varies in time. Moreover, the easiest way to derive the Hamiltonian and Lagrangian in angular coordinates is by first writing it down in Cartesian coordinates and then writing $x$ in terms of $q$. This difference in functional form is even more drastic in 3-dimensions where the Cartesian expression is identical, but the Hamiltonian and Lagrangian are substantially more complex. In Appendix F we derive additional examples showcasing the complexity difference between coordinates that implicitly enforce constraints and Cartesian coordinates. The constant mass matrix $M$ is in fact a general property of using Cartesian coordinates for these systems as shown in Section 6. By simplifying the functional form of $\mathcal{H}$ and $\mathcal{L}$, we make it easier for our models to learn.

## 5 Learning under explicit constraints

Although Cartesian coordinates reduce the functional complexity of the Hamiltonian and Lagrangian, they do not encode the constraints of the system. Therefore, we enforce the constraints explicitly for both Hamiltonian dynamics and Lagrangian dynamics using Lagrange multipliers.

**Hamiltonian mechanics with explicit constraints.** The dynamics of a Hamiltonian system can be derived by finding the stationary point of the action functional[2]

$$S[z] = \int \mathcal{L}(z(t)) dt = -\int \left[ \frac{1}{2} z(t)^\top J \dot{z}(t) + \mathcal{H}(z) \right] dt, \tag{3}$$

like in Lagrangian mechanics. Enforcing the necessary condition of a stationary point $\delta S = 0$ [3] yields the differential equation of the system $\dot{z} = J \nabla \mathcal{H}$ from Equation 1, which is shown in Appendix C.1. We can enforce constraints explicitly by turning this procedure into a constrained optimization problem via Lagrange multipliers.

Suppose we have $C$ holonomic[4] constraints $\{\Phi(x)_j = 0\}_{j=1}^C$ collected into a vector $\Phi(x) = 0$. We can differentiate the constraints to form an additional $C$ constraints that depend on the momentum $p$, since $0 = \dot{\Phi} = (D\Phi)\dot{x} = (D\Phi)\nabla_p\mathcal{H}$ where $D\Phi$ is the Jacobian of $\Phi$ with respect to $x$. If we collect $\Phi$ and $\dot{\Phi}$, we have $0 = \Psi(z) = (\Phi, \dot{\Phi}) \in \mathbb{R}^{2C}$ as the set of $2C$ constraints that we must enforce when finding a stationary point of $S$. We can enforce these constraints by augmenting the state $z$ with a vector of time dependent Lagrange multipliers $\lambda(t) \in \mathbb{R}^{2C}$, giving the augmented action

$$S[z, \lambda] = -\int \left[ \frac{1}{2} z^\top J \dot{z} + \mathcal{H}(z) + \Psi(z)^\top \lambda \right] dt. \tag{4}$$

Enforcing $\delta S = 0$ yields the differential equations that describe the state $z$ under explicit constraints $\Phi(x) = 0$:

$$\dot{z} = J\left[ \nabla\mathcal{H}(z) + (D\Psi(z))^\top \lambda \right], \tag{5}$$

where $D\Psi$ is the Jacobian of $\Psi$ with respect to $z$. Notice that each row $j$ of $D\Psi$ is the gradient of the constraint $\Psi(z)_j$ and is orthogonal to the constraint surface defined by $\Psi(z)_j = 0$. Left multiplying by $(D\Psi)$ to project the dynamics along these orthogonal directions gives $(D\Psi)\dot{z} = d\Psi/dt = 0$ which can then be used to solve for $\lambda$ to obtain $\lambda = -\left[ (D\Psi)J(D\Psi)^\top \right]^{-1}(D\Psi)J\nabla\mathcal{H}$. Defining the projection matrix $P := I - J(D\Psi)^\top \left[ (D\Psi)J(D\Psi)^\top \right]^{-1}(D\Psi)$, satisfying $P^2 = P$, the constrained dynamics of Equation 5 can be rewritten as

$$\boxed{\dot{z} = P(z)J\nabla\mathcal{H}(z)}. \tag{6}$$

Equation 6 can be interpreted as a projection of the original dynamics from Equation 1 onto the constraint surface defined by $\Psi(x) = 0$ in a manner consistent with the Hamiltonian structure.

**Lagrangian mechanics with explicit constraints.** We can perform a similar derivation for a constrained system with the Lagrangian $\mathcal{L}$. Given $C$ holonomic constraints $\Phi(x) = 0$, we show in Appendix C.1 that the constrained system is described by

$$\boxed{\ddot{x} = M^{-1}f - M^{-1}(D\Phi)^\top \left[ (D\Phi)M^{-1}(D\Phi)^\top \right]^{-1} \left[ (D\Phi)M^{-1}f + (D\dot{\Phi})\dot{x} \right]}, \tag{7}$$

where $D\Phi$ is the Jacobian of $\Phi$ with respect to $x$, $M = \nabla_{\dot{x}}\nabla_{\dot{x}}\mathcal{L}$ is the mass matrix, and $f = f_u + f_c$ is the sum of conservative forces and Coriolis-like forces. [5]

**Learning.** To learn $\mathcal{H}$ and $\mathcal{L}$, we parametrize them with a neural network and use Equation 6 and Equation 7 as the dynamics of the system. This approach assumes that we know the constraints $\Phi(x)$ and can compute their Jacobian matrices. Since mechanical systems in Cartesian coordinates have separable Hamiltonians and Lagrangian with constant $M$, our method can parametrize $M^{-1}$ with a learned positive semi-definite matrix instead of how it is usually done with a neural network [2, 14]. [6] For CHNN, we convert from $\dot{x}$ to $p$ and back using the learned mass matrix so that the model can be trained from regular position-velocity data.

## 6 Embedding 3D motion in Cartesian coordinates

Our goal is to learn in Cartesian coordinates, but how can we actually represent our systems in Cartesian coordinates? Point masses can be embedded in Cartesian coordinates by using a diagonal mass matrix, but it is less obvious how to represent extended rigid bodies like a spinning top. Below we show a general way of embedding rigid-body dynamics in an inertial frame while avoiding all non-Cartesian generalized coordinates such as Euler angles, quaternions, and axis angles which are commonly used in physics simulations [6, 19]. Additionally by avoiding quaternions, Euler angles, and cross products which are specialized to $\mathbb{R}^3$, we can use the same code for systems in any $\mathbb{R}^d$.

**Extended bodies in $d$-dimensions.** In Hamiltonian and Lagrangian mechanics, we may freely use any set of coordinates that describe the system as long as the constraints are either implicitly or

explicitly enforced. In fact, at the expense of additional constraints, any non-colinear set of $d$ points $\mathbf{x}_1, ..., \mathbf{x}_d$ of a rigid body in $\mathbb{R}^d$ are fixed in the body frame of the object and completely specify the orientation and center of mass of the object. The rigidity of the object then translates into distance constraints on these points.

Given an extended object with mass density $\rho$ that may be rotating and translating in space, coordinates in the body frame $\mathbf{y}$ and coordinates in the inertial frame $\mathbf{x}$ are related by $\mathbf{x} = R\mathbf{y} + \mathbf{x}_{cm}$, where $R$ is a rotation matrix and $\mathbf{x}_{cm}$ is the center of mass of the object in the inertial frame. As shown in Appendix C.2, the kinetic energy can be written in terms of $R$ and $\mathbf{x}_{cm}$ as

$$T = m\|\dot{\mathbf{x}}_{cm}\|^2/2 + m\operatorname{Tr}(\dot{R}\Sigma\dot{R}^\top)/2, \tag{8}$$

where $\Sigma = \mathbb{E}[\mathbf{y}\mathbf{y}^\top]$ is the covariance matrix of the mass distribution $\rho(\mathbf{y})$ in the body frame.

Collecting points $\{\mathbf{y}_i\}_{i=1}^d$ that are fixed in the body frame, we can solve $\mathbf{x}_i = R\mathbf{y}_i + \mathbf{x}_{cm}$ for $R$ to obtain the rotation matrix as a function of $\mathbf{x}_{cm}$ and $\{\mathbf{x}_i\}_{i=1}^d$. We may conveniently choose these $d$ points to be unit vectors aligned with the principal axes that form the eigenvectors of $\Sigma$ written in the inertial frame. As we show in Appendix C.2, when these principal axes and the center of mass are collected into a matrix $X = [\mathbf{x}_{cm}, \mathbf{x}_1, \ldots, \mathbf{x}_d] \in \mathbb{R}^{d\times(d+1)}$, we have $R = X\Delta$ where $\Delta = [-\mathbb{1}, I_{d\times d}]^T$. Plugging in $\dot{R} = \dot{X}\Delta$ into Equation 8 and collecting the $\dot{x}_{cm}$ term gives

$$T = \operatorname{Tr}(\dot{X}M\dot{X}^\top)/2 \qquad \text{where} \qquad M = m\begin{bmatrix} 1+\sum_i \lambda_i & -\lambda^\top \\ -\lambda & \operatorname{diag}(\lambda) \end{bmatrix}, \tag{9}$$

where the $\lambda = \operatorname{diag}(\Sigma)$ are the eigenvalues of $\Sigma$ are collected into a vector $\lambda$. Furthermore, $M^{-1} = m^{-1}\big(\mathbb{1}\mathbb{1}^\top + \operatorname{diag}([0, \lambda_1^{-1}, \ldots, \lambda_d^{-1}])\big)$. Finally, the mass matrix of multiple extended bodies is block diagonal where each block is of the form in Equation 9 corresponding to one of the bodies. Our framework can embed bodies of arbitrary dimension into Cartesian coordinates, yielding the primitives Obj0D, Obj1D, Obj2D, Obj3D corresponding to point masses, line masses, planar masses, and 3d objects.

**Rigidity Constraints.** To enforce the rigidity of the $d+1$ points that describe one of these extended bodies we use the distance constraints $\Phi(X)_{ij} = \|\mathbf{x}_i - \mathbf{x}_j\|^2 - \ell_{ij}^2 = 0$ on each pair. Technically $\ell_{ij} = 1$ for $i = 0$ or $j = 0$ and $\sqrt{2}$ otherwise, although these values are irrelevant. Given an Obj$d$D in $d$ ambient dimensions, this translates to $\binom{n+1}{2}$ internal constraints which are automatically populated for the body state $X \in \mathbb{R}^{d\times(n+1)}$.

**Joints between extended bodies.** For robotics applications we need to encode movable joints between two extended bodies. Given two bodies $A, B$ with coordinate matrices $X_A$ $X_B$, we use the superscripts $A, B$ on vectors to denote the body frame in which a given vector is expressed. A joint between two extended bodies $A$ and $B$ is specified by point at the pivot that can be written in the body frame of $A$ at location $\mathbf{c}^A$ and in the body frame of $B$ at location $\mathbf{c}^B$. Converting back to the global inertial frame, these points must be the same. This equality implies the linear constraint $\Phi(X_A, X_B) = X_A\tilde{c}^A - X_B\tilde{c}^B = 0$ where $\tilde{c} = \Delta\mathbf{c} + \mathbf{e}_0$. In Appendix C.2, we show how we incorporate axis restrictions, and how potentials and external forces can be expressed in terms of the embedded state $X$.

**Simple Links.** For the links between point masses (Obj0D) of pendulums, we use the simpler $\Phi(X_A, X_B) = \|X_A - X_B\|^2 - \ell_{AB}^2 = 0$ distance constraints on the $d$ dimensional state vectors $X_A$ and $X_B \in \mathbb{R}^{d\times 1}$. Since $P$ from Equation 6 depends only on $D\Phi$, the gradients of the constraints, *we need not know $\ell_{AB}$ in order to enforce these constraints, only the connectivity structure.*

**Summary.** To a learn a new system, we must specify a graph that lists the objects (of type Origin, Obj0D, Obj1D, Obj2D, Obj3D) and any constraint relationships between them (Link, Joint, Joint+Axis). These constraints are then converted into constraint Jacobians $D\Phi$, $D\Psi$, written out in Appendix D.4 which define the relevant projection matrices. For each body of type ObjND, we initialize a set of positive learnable parameters $m, \{\lambda_i\}_{i=1}^n$ which determine a mass matrix $M$ and $M^{-1}$ using Equation 9 and therefore the kinetic energy $T$. Combined with a neural network parametrizing the potential $V(X)$, these form a Hamiltonian $\mathcal{H}(z) = \mathcal{H}(X, P) = \operatorname{Tr}(PM^{-1}P^\top)/2 + V(X)$ or Lagrangian $\mathcal{L}(X, \dot{X}) = \operatorname{Tr}(\dot{X}M\dot{X}^\top)/2 - V(X)$, which could be augmented with additional terms to handle friction or controls as done in Zhong et al. [21]. Finally, $\dot{z} = P(z)J\nabla\mathcal{H}$ (Equation 6) and Equation 7 define the constrained Hamiltonian and Lagrangian dynamics that are integrated using a differentiable ODE solver.

# 7   Experiments

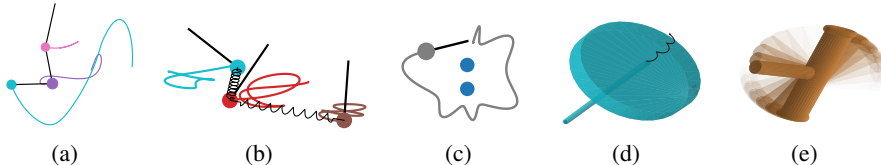

|  (a)  |  (b)  |  (c)  |  (d)  |  (e)  |

Figure 4: Systems with complicated dynamics that we simulate. In order from left to right: The N-pendulum, the 3-coupled-pendulum, the magnet pendulum, the gyroscope, and the rigid rotor.

**Datasets and simulated systems.** Previous work has considered relatively simple systems such as the 1 and 2-pendulum [3, 9], Cartpole, and Acrobot [2]. We extend the number of links and randomize both masses and joint lengths in the pendulum system to make it more challenging, shown in Figure 4(a). We also add four new benchmarks that test the ability to learn complicated trajectories in 3D. Figure 4(b) shows a sequence of 3 pendulums on ball joints that are coupled together with springs, Figure 4(c) shows a ball joint pendulum with a magnetic tip suspended above two repelling magnets with a complicated potential coming from the magnetic field, inducing chaotic behavior, Figure 4(d) shows a spinning top which exhibits both precession and nutation, and Figure 4(e) shows a free floating rigid rotor with unequal moments of inertia demonstrating the Dzhanibekov effect. Appendix E describes each system in detail and explains our data generation procedure.

**Training details.** Following [2, 18, 20] we minimize the error integrated over the trajectories. For each initial condition $(z_0, t_0)$ in a training minibatch corresponding to a true trajectory $((z_0, t_0), (z_1, t_1), \ldots, (z_n, t_n))$, the model predictions are rolled out with the ODE integrator $(\hat{z}_1, \hat{z}_2, ..., \hat{z}_n) = \text{ODESolve}(z_0, f_\theta, (t_1, t_2, ..., t_n))$ where $f_\theta$ is the learned dynamics. For each trajectory, we compute the $L_1$ loss averaged over each timestep of the trajectory[7] $L(z, \hat{z}) = \frac{1}{n} \sum_{i=1}^{n} \|\hat{z}_i - z_i\|_1$ and compute gradients by differentiating through ODESolve directly. We use $n = 4$ timesteps for our training trajectories and average $L(z, \hat{z})$ over a minibatch of size 200. To ensure a fair comparison, we first tune all models and then train them for 2000 epochs which was sufficient for all models to converge. For more details on training and settings, see Appendix D.2.

**Evaluating performance.** We evaluate the relative error between the model predicted trajectory $\hat{z}(t)$ and the ground truth trajectory $z(t)$ over timescales that are much longer than trajectories used at training. Our notion of relative error is $\text{Err}(t) = \|\hat{z}(t) - z(t)\|_2 / (\|\hat{z}(t)\|_2 + \|z(t)\|_2)$, which can be seen as a bounded version of the usual notion of relative error $\|\hat{z}(t) - z(t)\|_2 / \|z(t)\|_2$. $\text{Err}(t)$ measures error independent of the scale of the data and approaches 1 as predictions become orthogonal to the ground truth or $\|\hat{z}\| \gg \|z\|$. Since the error in forward prediction compounds multiplicatively, we summarize the performance over time by the geometric mean of the relative error over that time interval. The geometric mean of a continuous function $h$ from $t = 0$ to $t = T$ is $\bar{h} = \exp(\int_0^T \log h(t) dt / T)$, which we compute numerically using the trapezoid rule. We compare our Constrained Hamiltonian and Lagrangian Neural Networks (CHNNs, CLNNs) against Neural-ODEs [1], Hamiltonian Neural Networks (HNNs) [9], and Deep Lagrangian Networks (DeLaNs) [14] on the systems described above. We also evaluate the models' abilities to conserve the energy of each system in Appendix B.1 by plotting the relative error of the trajectories' energies.

**Performance on $N$-pendulum systems.** The dynamics of the $N$-pendulum system becomes progressively more complex and chaotic as $N$ increases. For each model, we show its relative error over time averaged over the $N_{test} = 100$ initial conditions from the test set in Figure 5 with $N = 1, 2, 3, 5$. Note that each training trajectory within a minibatch for the $N$-pendulum systems is only $T_{minibatch} = 0.12s$ long whereas Figure 5 evaluates the models for $3s$. All models perform progressively worse as $N$ increases, but CHNN and CLNN consistently outperform the competing methods with an increasing gap in the relative error as $N$ increases and the dynamics become increasingly complex. We present Figure 5 in linear scale in Appendix B, emphasizing that CHNN and CLNN have lower variance than the other methods.

Figure 6 left shows the quality of predictions on the 2-pendulum over a long time horizon of $15s$ with the $y$ coordinate of the second mass for a given initial condition. As the trajectories for $N$-

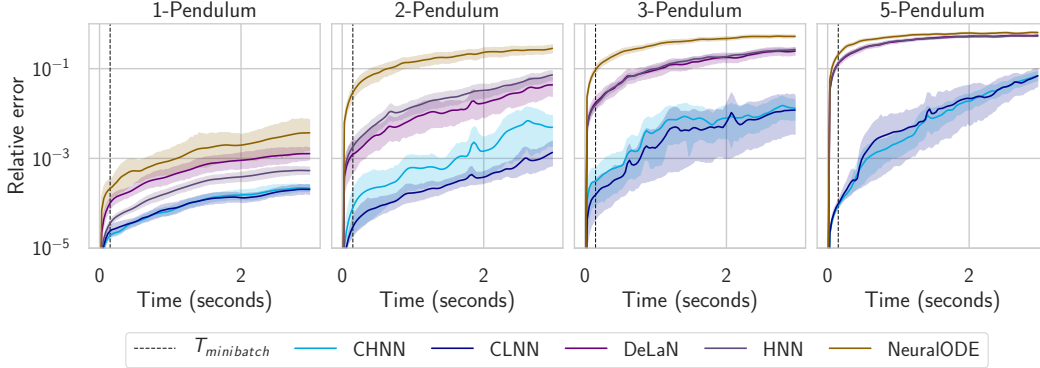

Figure 5: The relative error in the state for rollouts of the baseline NN, HNN, LNN models compared to CHNN and CLNN on the Pendulum Chain tasks. Curves are averaged over $N_{test} = 100$ initial conditions and shaded regions are 95% confidence intervals. The vertical axis is log-scaled, meaning that CHNN and CLNN actually have lower variance than the other models. We show this figure in linear scale in Appendix B.

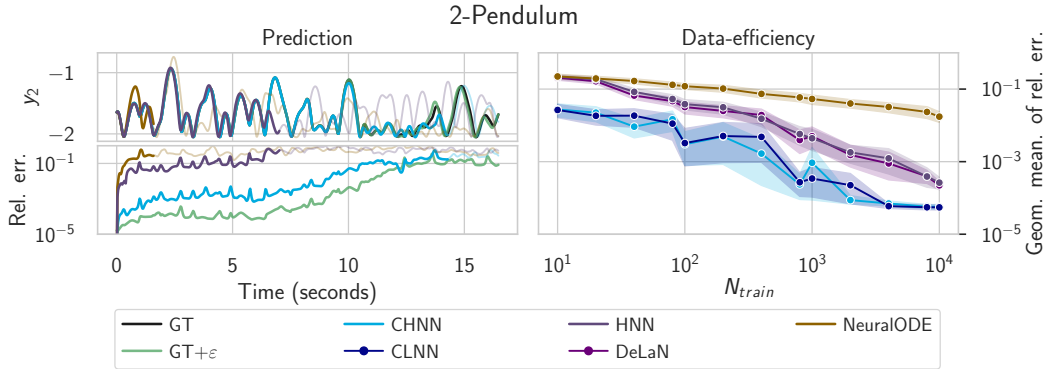

Figure 6: **Left:** The rollout predictions for the $y$ coordinate of the second bob of a 2-pendulum and relative error over an extended timespan foor CHNN and HNN. Trajectories are faded out several steps after reaching 50% relative error. As the dynamics are chaotic, we also plot a ground truth trajectory that has been perturbed by $\varepsilon = 10^{-5}$ showing the natural chaotic growth of error. **Right:** CHNN and CLNN can achieve the same performance with significantly less data than the baselines. Curves are averaged over $N_{test} = 100$ initial conditions and shaded regions are 95% confidence intervals.

pendulum are chaotic for $N \geq 2$, small errors amplify exponentially. Even a small perturbation of the initial conditions integrated forward with the ground truth dynamics leads to noticeable errors after $15s$. Notably, our models produce accurate predictions over longer timespans, generalizing well beyond the training trajectory length of $T_{minibatch} = 0.12$ seconds.

**Data-efficiency.** As shown in Section 4, the analytic form of the Hamiltonian, and the Lagrangian, are overwhelmingly simpler in Cartesian coordinates. Intuitively, simpler functions are simpler to learn, which suggests that our explicitly-constrained models should be more data-efficient. Figure 6 right compares the data-efficiency of the models on the chaotic 2-pendulum task. We choose this system because it has been evaluated on in prior work [3], and it is one that previous models can still learn effectively on. Our CHNN and CLNN models achieve a lower geometric average error rate using $N_{train} = 20$ trajectories than HNN with $N_{train} = 200$ trajectories and NeuralODE with $N_{train} = 1600$ trajectories.

**Performance on 3D systems.** The pure intrinsic constraint generalized coordinate approach for 3D systems must rely on spherical coordinates and Euler angles, which suffer from coordinate singularities such as gimbal lock. These coordinate singularities lead to singular mass matrices which can destabilize training. We had to apply rotated coordinate systems to avoid singularities when training the baselines models as decribed in Appendix E. In contrast, CHNN and CLNN naturally circumvent these problems by embedding the system into Cartesian coordinates. As shown in Figure 7, CHNN and CLNN outperform the competing methods both on the tasks where the complexity derives primarily from the coordinate system, the Gyroscope and the Rigid rotor, and on the tasks

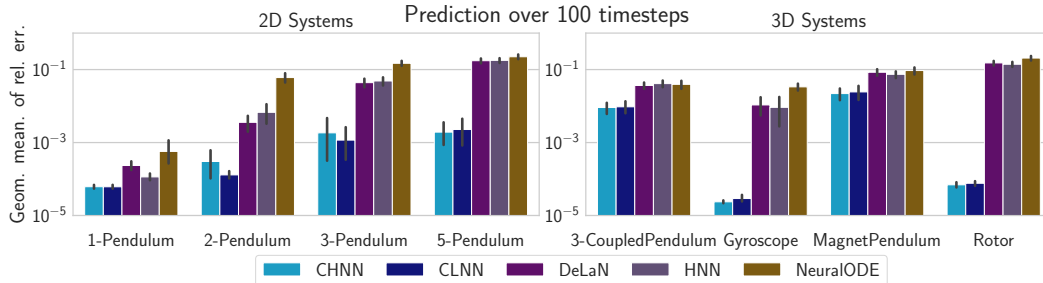

Figure 7: The geometric mean of relative error averaged over $N_{test} = 100$ initial conditions with 95% confidence interval error bars. Models with explicit constraints outperform competing methods on all systems. **Left**: The 2D $N$-pendulum systems which are chaotic for $N \geq 2$. **Right**: The 3D systems of which the spring-coupled pendulum and the Magnet-Pendulum are chaotic.

where the complexity comes from the potential: the spring coupling in 3-CoupledPendulum and the magnetic field in MagnetPendulum.

# 8 Conclusion

We have demonstrated that Cartesian coordinates combined with explicit constraints make the Hamiltonians and Lagrangians of physical systems easier to learn, improving the data-efficiency and trajectory prediction accuracy by two orders of magnitude. We have also shown how to embed arbitrary extended body systems into purely Cartesian coordinates. As such, our approach is applicable to rigid body systems where the state is fully observed in a 3D space, such as in robotics. However, Cartesian coordinates are only possible for systems in physical space, which precludes our method from simplifying learning in some Hamiltonian systems like the Lotka-Volterra equations.

There are many exciting directions for future work. Our approach is compatible with recent works that learn dynamical systems with controls [14, 20] and in the presence of dissipative forces [21]. While we develop the method in a continuous time, there are circumstances involving collision, contacts, and friction where discrete time would be advantageous. Although we used the explicit constraint framework only for Cartesian coordinates in this paper, they can also enforce additional constraints in generalized coordinates, allowing us to pick the best coordinate system for the job. We hope that this approach can inspire handling other kinds of constraints such as gauge constraints in modeling electromagnetism.Finally, although our method requires the constraints to be known, it may be possible to model the constraints with neural networks and propagate gradients through the Jacobian matrices to learn the constraints directly from data.

# 9 Broader Impacts

Being able to model physical systems accurately has broad applications in robotics, model-based reinforcement learning, and data-driven control systems. A model that can learn the dynamics of arbitrary systems would greatly reduce the amount of expert-time needed to design safe and accurate controllers in a new environment. Although we believe that there are many advantages for using generic neural networks in robotics and control over traditional expert-in the-loop modeling and system identification, neural network models are harder to interpret and can lead to surprising and hard-to-understand failure cases. The adoption of neural network dynamics models in real world control and robotics systems will come with new challenges and may not be suitable for critical systems until we better understand their limitations.

**Acknowledgements.** This research is supported by an Amazon Research Award, Facebook Research, Amazon Machine Learning Research Award, NSF I-DISRE 193471, NIH R01 DA048764-01A1, NSF IIS-1910266, and NSF 1922658 NRT-HDR: FUTURE Foundations, Translation, and Responsibility for Data Science.

## Footnotes

[2]Which is in fact exactly the Lagrangian action of the original system, see Appendix C.1 for more details.

[3]$\delta S$ is the variation of the action with respect to $z$, using the calculus of variations.

[4]Holonomic constraints are equations that give the relationship between position coordinates in the system.

[5]$f_u(x, \dot{x}) = \nabla_x\mathcal{L}$ and $f_c(x, \dot{x}) = -(\nabla_{\dot{x}}\nabla_x\mathcal{L})\dot{x}$, but $f_c = 0$ in Cartesian coordinates.

[6]To enforce positive semi-definiteness of $M^{-1}$, we parametrize the Cholesky decomposition of the matrix $M^{-1}$. In practice, we use a more specialized parametrization of $M$ that is block diagonal but still fully general even when we do not know the ground truth Hamiltonian or Lagrangian, shown in Equation 9

[7]We found that the increased robustness of $L_1$ to outliers was beneficial for systems with complex behavior.

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
