[Supplementary Material 1]

# Appendices

## A Overview

Appendix B presents additional experimental results of the paper. In Appendix B.1 we compare how well the predicted trajectories conserve energy as a function of time. In Appendix B.2 we show the relative error on the $N$-Pendulum systems on a linear scale, which emphasizes that trajectories predicted by CHNNs and CLNNs have lower variance. In Appendix B.3 we quantify constraint drift and passively enforced constraints. In Appendix B.4 we investigate the complexity of the learning task through the lens of effective dimension [16].

Appendix C presents the derivation of constrained Hamiltonian and Lagrangian mechanics, and the derivations used to embed 3D motion in Cartesian coordinates.

Appendix D presents the implementation and training details for our method and the baselines, including documentation of the constraint Jacobians in Appendix D.4.

Appendix E details the 3D systems that make up the new benchmark datasets, including their Hamiltonians.

Finally, Appendix F further demonstrates the complexity of generalized coordinates by deriving the Hamiltonians in generalized coordinates for each of these systems.

## B Additional results

### B.1 Energy Conservation

When models approximate the true Hamiltonian or the true Lagrangian, they are able to approximately conserve energy. We compare conservation of the true energy of the system of trajectories predicted by each model over time in Figure 8, showing the relative error $|\mathcal{H}(z) - \mathcal{H}(\hat{z})| / (|\mathcal{H}(z)| + |\mathcal{H}(\hat{z})|)$. CHNNs and CLNNs outperform Neural ODEs, HNNs, and DeLaNs on all systems. All models are able to approximately conserve the true energy of the system on the simple 1-pendulum task but the energy of the trajectories predicted by the baseline models quickly diverge as we transition to the chaotic 3-pendulum and 5-pendulum systems.

Figure 8: Absolute relative error between the true energy of the predicted trajectories given initial condition $z_0$ and the true energy of the ground truth trajectories starting at $z_0$. Curves are averaged over $N_{test} = 100$ initial conditions and shaded regions are 95% confidence intervals.

Figure 9: Here we show the linear scale version of Figure 5 which better visually demonstrates the performance difference between our method and the baseline comparisons. The 95% confidence interval is only perceptively larger for CHNNs and CLNNs in log-scale because they have lower relative error. As shown here, CHNNs and CLNNs in fact have lower variance than the baseline models.

## B.2   Relative error in linear scale

We show the relative error of each model on the pendulum systems in linear scale in Figure 9. We presented the same data in log scale in the main text. By presenting the relative errors in linear scale, it is more visually obvious that CHNNs and CLNNs have lower variance in error than the other methods.

## B.3   Removed constraints and constraint violation.

Our method uses Lagrange multipliers to enforce constraints since Cartesian coordinates do not enforce constraints by themselves. In Figure 10(a), we verify that performance rapidly degrades when these constraints are not explicitly enforced for CHNN and CLNNs. In principle the learned Hamiltonian of the model can compensate by approximately enforcing constraints such as with springs, however in practice this does not work well. We compare the performance of our CHNN and CLNN models on the 3-Pendulum (still in Cartesian coordinates) but where the distance constraints are successively removed starting from the bottom. While performance of the base models are >100 times better than HNN, removing even a single constraint results in performance that is slightly worse than an HNN. This experiment shows that both Cartesian coordinates and the explicit enforcement of constraints are necessary for the good performance.

Although the continuous time dynamics of Equation 6 and Equation 7 exactly preserve the constraints, numerical integration can cause small drifts in the violation of these constraints. The amount of violation can be controlled by the tolerance on the integrator, and we found it to be only a small contribution to the rollout error. Symplectic methods for constrained Hamiltonian systems where both phase space area (the symplectic form) and the constraints are exactly preserved by the discrete step integrator have been developed in the literature [11] and may prove helpful for very long rollouts. Figure 10(b) shows the amount of constraint violation (measured by RMSE on $\Phi$) over time of our models as they predict forward in time. Figure 10(c) shows the amount of constraint violating as a function of the integrator relative tolerance. As expected, the amount of violation decreases as we make the tolerance more strict.

## B.4   Effective Dimension

We empirically evaluate the complexity of the parameterized Hamiltonian $\mathcal{H}_\theta$ learned by CHNN and HNN using effective dimensionality [15, 16]. The effective dimensionality of parameterized model $\mathcal{H}_\theta$ is given by

$$\text{ED}(\mathcal{H}_\theta) = \sum_{i=1}^{L} \frac{\lambda_i}{\lambda_i + z} \tag{10}$$

where $z$ is a soft cutoff hyperparameter and $\lambda_i$ are eigenvalues of the Hessian of the loss function with respect to the parameters $\theta$. Effective dimensionality characterizes the complexity of a model

(a)          (b)          (c)

Figure 10: **Left:** We sequentially remove the enforced constraints from the CHNN and CLNN models for the 3-pendulum and evaluate the geometric mean of the relative error against that of a HNN which has all constraints enforced implicitly. Performance degrades rapidly showing that embedding into Cartesian coordinates without explicit constraints is insufficient. **Middle:** The figure shows how constraint drift leads to a small but increasing constraint violation over time (at $\mathrm{rtol} = 10^{-6}$). **Right:** The geometric mean of the violation over the trajectory is plotted against the relative tolerance of the integrator. By adjusting the tolerance the violation can be controlled.

Figure 11: Normalized eigenvalues $\lambda_i / \sum_{i=1}^{L} \lambda_j$ for the top $N = 500$ eigenvalues. Shaded region is 95% confidence interval averaged over 3 independently trained models. The eigenvalues of CHNN decays more rapidly than those of HNN as the tasks become more difficult, corresponding to a lower effective dimensionality.

by the decay of its eigenspectrum. Small eigenvalues with $\lambda_i \ll z$ do not contribute to $\mathrm{ED}(\mathcal{H}_\theta)$, and eigenvalues above the soft threshold set by $z$ contribute approximately 1 to $\mathrm{ED}(\mathcal{H}_\theta)$. More complex models have higher effective dimensionality corresponding to slower eigenvalue decay with more eigenvalues greater than the threshold $z$.

The loss function differs in scale between CHNN and HNN, with one computing the $L_1$ error between Cartesian coordinates and the other computing the $L_1$ error between angular coordinates, which affects the scale of the eigenvalues. Thus we normalize each eigenvalue by $\sum_{i=1}^{L} \lambda_i$ before computing the effective dimensionality. In otherwords, we compute effective dimensionality using $\lambda_i \leftarrow \lambda_i / \sum_{j=1}^{L} \lambda_j$. Figure 11 shows the eigenspectra of CHNN and HNN on the $N$-Pendulum systems averaged over 3 runs. Since the models have 133,636 and 269,322 parameters respectively, we only compute the top 500 eigenvalues using the Lanczos algorithm. Based on the elbow in the eigenspectra in Figure 11, we set $z = 10^{-3}$ when computing the effective dimensionality shown in Figure 12. Figure 12 shows that the effective dimensionality of HNNs are higher than that of CHNNs which may correspond to the more complex Hamiltonian that HNNs must learn.

## C  Supporting derivations

### C.1  Constrained Hamiltonian and Lagrangian Mechanics

**Derivation of Hamiltonian mechanics.** Let $z = (x, p)$ and assume that the Hamiltonian $\mathcal{H}$ has no explicit dependence on time. The true trajectory $z$ of a Hamiltonian system is a stationary point of the action functional

$$S[z] = -\int \left[ \frac{1}{2} z(t)^\top J \dot{z}(t) + \mathcal{H}(z) \right] dt. \qquad (11)$$

Figure 12: Effective dimensionality of CHNN and HNN on the $N$-Pendulum tasks with $z = 10^{-3}$ computed using normalized eigenvalues. Shaded region is 95% confidence interval averaged over 3 independently trained models.

Finding the stationary point by varying the action $\delta S = 0$, one recovers the Hamiltonian equations of motion $\dot{z} = J\nabla\mathcal{H}$ [4]:

$$
\begin{aligned}
\delta S &= -\int \left[\frac{1}{2}\delta z^\top J\dot{z} + \frac{1}{2}z^\top J\delta\dot{z} + \delta\mathcal{H}(z)\right]dt \\
&= -\int \left[\frac{1}{2}\delta z^\top (J - J^\top)\dot{z} + \delta z^\top \nabla\mathcal{H}\right]dt \\
&= -\int \delta z^\top \left[J\dot{z} + \nabla\mathcal{H}\right]dt = 0 \\
J\dot{z} &= -\nabla\mathcal{H} \\
\dot{z} &= J\nabla\mathcal{H}
\end{aligned}
\tag{12}
$$

where we use $J = J^\top$ and integrate by parts in Equation 12. We use $J^2 = -I$ to obtain the final equation. In fact, the action is the exact same as the one used to derive the Lagrangian equations of motion. Splitting apart $z = (q, p)$, we have $S = \int \frac{1}{2}(p^\top \dot{q} - q^\top \dot{p}) - H(q, p)dt$ and integrating by parts $S = \int p^\top \dot{q} - H(q, p)dt = \int \mathcal{L}(q, \dot{q})dt$. See the main text for how this approach generalizes in the case of constrained Hamiltonian mechanics.

**Derivation of constrained Lagrangian mechanics.** We can follow a similar derivation for the constrained Lagrangian formalism. Here, the state is $z = (x, \dot{x})$ for a Lagrangian $\mathcal{L}(z(t)) = \mathcal{L}(q(t), \dot{q}(t))$. Although the constrained Lagrangian formalism allows for $\dot{q}$ dependent constraints, known as pfaffian constraints, we will assume a set of $m$ holonomic constraints $\Phi(q)$ for simplicity. Like before, since the constraints are not implicit in the coordinate choice, we must enforce them explicitly using Lagrange multipliers. Given a set of $m$ holonomic constraints on the position $\Phi(x)_a = 0$ for $a = 1, 2, ..., m$ collected as a column vector $\Phi(x) = 0$, one adds a set of time dependent Lagrange multipliers $\lambda_a(t)$ collected as a column vector $\lambda(t)$ to the state with the augmented action:

$$
S[z, \lambda] = \int \left[\mathcal{L}(x, \dot{x}) - \Phi(x)^\top \lambda(t)\right]dt.
\tag{13}
$$

Enforcing $\delta S/\delta x = 0$ and $\delta S/\delta\lambda = 0$ with vanishing boundary conditions gives $\Phi(x) = 0$ and

$$
\ddot{x} = M^{-1}[f - (D\Phi)^\top \lambda]
\tag{14}
$$

where $M = \nabla_{\dot{x}}\nabla_{\dot{x}}\mathcal{L}$ is the mass matrix and $f = f_u + f_c$ is the sum of conservative forces $f_u(x, \dot{x}) = \nabla_x\mathcal{L}$ and Coriolis-like forces $f_c(x, \dot{x}) = -(\nabla_{\dot{x}}\nabla_x\mathcal{L})\dot{x}$ which vanish in Cartesian coordinates. Here $D\Phi$ is the Jacobian of $\Phi$ with respect to $x$.

In order to solve for $\lambda$, we can multiply Equation 14 on the left by $D\Phi$ to get:

$$
(D\Phi)\ddot{x} = (D\Phi)M^{-1}f - (D\Phi)M^{-1}(D\Phi)^\top \lambda.
\tag{15}
$$

Since the constraints are preserved, $d\Phi/dt = (D\Phi)\dot{x} = 0$, and taking time derivatives a second time gives $(D\Phi)\ddot{x} = -(D\dot{\Phi})\dot{x}$ which we can substitute into Equation 15. Rearranging terms and

inverting the matrix $\left[(D\Phi)M^{-1}(D\Phi)^\top\right]$, we get the solution for $\lambda$:

$$\lambda = \left[(D\Phi)M^{-1}(D\Phi)^\top\right]^{-1}\left[(D\Phi)M^{-1}f + (D\dot{\Phi})\dot{x}\right]. \tag{16}$$

Combining equations Equation 16 and Equation 14 then gives the equations of motion in the constrained Lagrangian formalism. Notably, this derivation holds not just for Cartesian coordinates but in any coordinate system. The difference is that in other coordinate systems, the mass matrix may not be constant $M(x,\dot{x})$ and the Coriolis term $f_c(x,\dot{x})$ may be nonzero. Additionally, the derivation is not specialized to mechanical systems with quadratic kinetic energies, but also applies to more general Lagrangians such as in electrodynamics or special relativity as explored in Cranmer et al. [3].

## C.2 3D systems derivation

Given an extended object with mass density $\rho$ where the coordinates of points in the body frame $y$ are related to the coordinates in the inertial frame $\mathbf{x}$ by $\mathbf{x} = R\mathbf{y} + \mathbf{x}_{cm}$, we can split up the kinetic energy into a translational component depending on $\dot{\mathbf{x}}_{cm}$ and a rotational component depending on $\dot{R}$. The mass and center of mass of the object are given by $m = \int d\rho$, $x_{cm} = (1/m)\int x d\rho(x)$. The kinetic energy of the body is then:

$$T = (1/2)\int \dot{\mathbf{x}}^\top \dot{\mathbf{x}} d\rho(x) = (1/2)\int (\|\dot{\mathbf{x}}_{cm}\|^2 + 2\dot{\mathbf{x}}_{cm}^\top \dot{R}\mathbf{y}_i + \|\dot{R}\mathbf{y}_i\|^2)d\rho \tag{17}$$

$$= (1/2)m\|\dot{\mathbf{x}}_{cm}\|^2 + (1/2)\int \mathrm{Tr}(\mathbf{y}\mathbf{y}^\top \dot{R}^\top \dot{R})d\rho(y) \tag{18}$$

$$= (1/2)m\|\dot{\mathbf{x}}_{cm}\|^2 + (1/2)m\,\mathrm{Tr}(\dot{R}\Sigma\dot{R}^\top) = T_{trans} + T_{rot} \tag{19}$$

where we have defined the matrix of second moments in the body frame $\Sigma = \mathbb{E}[\mathbf{y}\mathbf{y}^\top] = \frac{1}{m}\int \mathbf{y}\mathbf{y}^\top d\rho(y)$, the covariance matrix of the mass distribution, which is a constant. The middle term in Equation 17 vanishes since $\mathbb{E}[\mathbf{y}] = (1/m)\int \mathbf{y}d\rho(y) = 0$. This decomposition is exactly mirrors the usual decomposition into rotational energy $T_{rot} = \omega^\top \mathcal{I}\omega$ but is written out differently to avoid angles and specialization to 3D. In 3D, the angular velocity is $\omega = *(\dot{R}R^{-1})$ where $*$ pulls out the components above the diagonal, and the inertia matrix is related to $\Sigma$ by $\mathcal{I} = m(\mathrm{Tr}(\Sigma)I_{3\times 3} - \Sigma)$.

Since the configuration space of a rigid body in $d$ dimensions is SE(3), we can embed it in a Cartesian space by choosing any $d$ linearly independent points $\{\mathbf{y}_i\}_{i=1}^d$ that are fixed in the body frame and for convenience an additional point for the center of mass $\mathbf{y}_{cm} = 0$. We may choose these vectors to lie along the principal axis of the object: $\mathbf{y}_i = Q_i$ from the eigen-decomposition $\Sigma = Q\Lambda Q^\top$. Collecting these points expressed in the body frame into a matrix $X = [\mathbf{x}_{cm}, \mathbf{x}_1, \ldots, \mathbf{x}_d] \in \mathbb{R}^{d\times(d+1)}$, we can express the constraint relating the two frames $R\mathbf{y} = \mathbf{x} - \mathbf{x}_{cm}$ as a linear system:

$$RQ = X\begin{bmatrix}-\mathbb{1}^\top \\ I\end{bmatrix} = X\Delta$$
$$R = X\Delta Q^\top$$
$$\dot{R} = \dot{X}\Delta Q^\top$$

for the matrix $\Delta = [-\mathbb{1}, I]^\top$. Combining with Equation 17, the kinetic energy can be rewritten as

$$T = m\|\dot{\mathbf{x}}_{cm}\|^2/2 + m\,\mathrm{Tr}(\dot{X}\Delta[Q^\top \Sigma Q]\Delta^\top \dot{X}^\top)/2$$
$$T = m\,\mathrm{Tr}(\dot{X}[\mathbf{e}_0\mathbf{e}_0^\top + \Delta\Lambda\Delta^\top]\dot{X}^\top)/2,$$

where we have made use of $Q^\top \Sigma Q = Q^\top Q\Lambda Q^\top Q = \Lambda$. Or alternatively, we can choose the coordinate system in the body frame so that $Q = I$. Defining $\Lambda = \mathrm{diag}(\lambda)$ we can collect terms into a single matrix $M$:

$$T(X) = \mathrm{Tr}(\dot{X}M\dot{X}^\top)/2 \qquad \text{where} \qquad M = m\begin{bmatrix}1 + \sum_i \lambda_i & -\lambda^\top \\ -\lambda & \mathrm{diag}(\lambda)\end{bmatrix}. \tag{20}$$

**Forces on extended bodies.** Given a point with components $\mathbf{c} \in \mathbb{R}^d$ in the body frame, the vector $x_c$ in the inertial frame has components

$$\mathbf{x}_c = \mathbf{x}_{cm} + \sum_i \mathbf{c}_i(\mathbf{x}_i - \mathbf{x}_{cm}) = X[\mathbf{e}_0 + \Delta \mathbf{c}] = X\tilde{c},$$

where $\tilde{c} = \mathbf{e}_0 + \Delta \mathbf{c}$. Forces $\mathbf{f} \in \mathbb{R}^d$ that are applied at that location yield generalized forces on the point collection $F_{k\alpha} = f_k \tilde{c}_\alpha$ for $k = 1, 2, ..., d$ and $\alpha = 0, 1..., d$ (in the sense that the unconstrained equations of motion would be $F = \ddot{X}M$). To see this, consider a potential that depends on the location of a certain point $\mathbf{x}_c$ of the rigid body $V = V(\mathbf{x}_c) = V(X\tilde{c})$. Potentials depending on the $\mathbf{x}_c$ can be expressed by simply substituting $X\tilde{c}$ in for $x_c$ in the form of the potential. The (generalized) forces can then be derived via chain rule: $F_{k\alpha} = \sum_j \frac{\partial V(\mathbf{x}_c)}{\partial (\mathbf{x}_c)_j} \frac{\partial (\mathbf{x}_c)_j}{\partial X_{k\alpha}} = f_k \tilde{c}_\alpha$.

**Rotational axis restrictions.** Not all joints allow free rotation in all dimensions. Instead it may be that a joint connecting bodies $A, B$ can only rotate about a single axis $\mathbf{u}$. In this case, the axis $\mathbf{u}$ is fixed in the two frames, and is related by a fixed change of basis when expressed in the body frame of $A$ and the body frame of $B$. This setup gives the constraint $\Phi(X_A, X_B) = R_A \mathbf{u}^A - R_B \mathbf{u}^B = X_A \Delta \mathbf{u}^A - X_B \Delta \mathbf{u}^B = 0$, since $R_A = X_A \Delta$, similar to the joint constraint but without the extra $\mathbf{e}_0$.

# D  Implementation details

## D.1  Dataset generation

We generate synthetically datasets using by plugging in known Hamiltonians from a variety of test systems to the framework described above. For each experiment and each system in Figure 5, Figure 7, Figure 8, and Figure 9, we create a training set by sampling $N_{train} = 800$ initial conditions and evaluate the dynamics at 100 timesteps separated by $\Delta t$ specific to the system. We integrate the dynamics using an adaptive ODE solver, Runge-Kutta4(5), implemented by Chen et al. [1] with a relative tolerance of $10^{-7}$ and an absolute tolerance of $10^{-9}$. We divide each trajectory into 20 non-overlapping chunks each with 5 timesteps. Finally, we choose a chunk at random from each trajectory, resulting in an aggregate training set of $N_{train} = 800$ trajectories each with 4 timesteps. At training time, we create a minibatch by sampling $m = 200$ of these shortened trajectories. We also create a separate set of $N_{test} = 100$ trajectories following the above procedure except that each test trajectory contains the full 100 timesteps without chunking.

For the data-efficiency experiments in Figure 1 (Middle) and Figure 6, we generate $N = 10000$ training trajectories each with 4 timesteps following the same procedure as the previous paragraph. We then choose the first $N_{train}$ trajectories among these $N$ trajectories as we vary $N_{train}$ on the x-axis. This ensure that the the sequence of training trajectories for each $N_{train}$ is a sequence of monotone increasing sets as $N_{train}$ increases.

## D.2  Architecture

For the baseline models which are trained on angular data, care must be taken to avoid discontinuities in the training data. In particular, we unwrap $(\pi - \epsilon \to -\pi + \epsilon)$ to $(-\infty, \infty)$ for the integration and embed each angle $\theta$ into $\sin\theta, \cos\theta$ before passing into the network as in Zhong et al. [20] and Cranmer et al. [3], to improve generalization. For all neural networks, we use 3 hidden layers each with 256 hidden units and $\tanh$ activations. We use this architecture to parametrize the potential energy $V$ for all models. We also use this architecture to parametrize the dynamics for the baseline Neural ODE.

The kinetic energy term $p^\top M(q)^{-1} p/2 = \dot{q}^\top M(q)\dot{q}/2$ is handled differently between our explicitly constrained models and the baselines. For the baseline HNNs and DeLaNs, we use the above neural network to parameterize the lower triangular matrix $L(q)$ from the Cholesky decomposition $L(q)L(q)^\top = M^{-1}(q)$, as done in Lutter et al. [14] and Zhong et al. [20] since the mass matrix in general is a function of $q$. For CHNNs and CLNNs we parameterize $M$ and $M^{-1}$ using learned parameters $m, \{\lambda_i\}_{i=1}^d$ as described in Equation 9 since $M$ and $M^{-1}$ are constant in Cartesian coordinates.

### D.3 Model selection and training details

We tune all models on the 3-Pendulum system, which is of intermediate difficulty, using the integrated trajectory loss evaluated on a separate validation set of 100 trajectories. We find that using AdamW [13] with a learning rate of $3 \times 10^{-3}$ and weight decay of $10^{-4}$ along with learning rate cosine annealing [12] without restarts generally works the best for all models across systems. To ensure convergence of all models, we train all models for 2000 epochs even though CHNN and CLNN usually converge within a few hundred epochs. With the exception of the magnetic pendulum and the rigid rotor, which require a lower learning rate and fewer epochs, we use these settings for all experiments. Despite our best efforts to circumvent this issue, HNN and DeLaN encounter gimbal lock for 3D systems due to their choice of angular coordinates, which causes the loss function to explode if trained for too long. Thus we could only train HNN and DeLaN for 200 epochs on the rigid rotor, which was empirically sufficient to flatten the training loss. We circumvent the coordinate singularity for these baseline models by rotating the coordinate system by $\pi/2$ so that the straight up/straight down configurations do not correspond to a coordinate singularity.

### D.4 Constraint Jacobians

**Distance constraints** For each of the position constraints $\Phi_{ij} = \|\mathbf{x}_i - \mathbf{x}_j\|^2$, there is a conjugate constraint on the velocity:

$$\dot{\phi}_{ij} = 2(\mathbf{x}_i - \mathbf{x}_j) \cdot (\dot{\mathbf{x}}_i - \dot{\mathbf{x}}_j) = 0 \tag{21}$$

Collecting the constraints: $\Phi = [\phi, \dot{\phi}]$ and taking derivatives with respect to $x_{k\ell}$ and $\dot{x}_{k\ell}$, the Jacobian matrix $D_{x,\dot{x}}\Phi \in \mathbb{R}^{2nd \times 2E}$ takes the simple form:

$$(D\Phi)_{(k\ell)(nm)} = \begin{bmatrix} \frac{\partial \phi_{nm}}{\partial x_{kl}} & \frac{\partial \dot{\phi}_{nm}}{\partial x_{kl}} \\ \frac{\partial \phi_{nm}}{\partial \dot{x}_{kl}} & \frac{\partial \dot{\phi}_{nm}}{\partial \dot{x}_{kl}} \end{bmatrix} = \begin{bmatrix} 2(x_{kn} - x_{km})(\delta_{n\ell} - \delta_{m\ell}) & 2(\dot{x}_{kn} - \dot{x}_{km})(\delta_{n\ell} - \delta_{m\ell}) \\ 0 & 2(x_{kn} - x_{km})(\delta_{n\ell} - \delta_{m\ell}) \end{bmatrix}$$

In $\dot{x}$ is related to $p$ by $\dot{x}_{k\ell} = \frac{\partial \mathcal{H}}{\partial p_{k\ell}}$ which in general could be a nonlinear function of $x, p$; however, in mechanics $\dot{x}_{k\ell} = \sum_n p_{kn} M_{\ell n}^{-1}$ or in matrix form $\dot{X} = PM^{-1}$. This allows relating the derivatives: $\frac{\partial \Phi}{\partial p}_{n\ell} = \sum_n M_{\ell n}^{-1} \frac{\partial \Phi}{\partial \dot{x}}_{kn}$. In matrix form: $D_p \Phi = (I_{3\times 3} \otimes M^{-1})D_{\dot{x}}\Phi$.

**Joint constraints**

For a joint constraint $\Phi(X_A, X_B) = X_A \tilde{c}^A - X_B \tilde{c}^B$ connecting bodies $A$ and $B$, there is a similar constraint on the velocities. The Jacobian matrix $D_{x,\dot{x}}\Phi \in \mathbb{R}^{2nd(d+1) \times 2d}$ takes the form:

$$(D\Phi)_{(k\alpha n)(i)} = \begin{bmatrix} \frac{\partial \phi_{ij}}{\partial x_{k\alpha}^n} & \frac{\partial \dot{\phi}_{ij}}{\partial x_{k\alpha}^n} \\ \frac{\partial \phi_{ij}}{\partial \dot{x}_{k\alpha}^n} & \frac{\partial \phi_{ij}}{\partial \dot{x}_{k\alpha}^n} \end{bmatrix} = \begin{bmatrix} \tilde{c}_\alpha^A \delta_{ki}\delta_{nA} - \tilde{c}_\alpha^B \delta_{ki}\delta_{nB} & 0 \\ 0 & \tilde{c}_\alpha^A \delta_{ki}\delta_{nA} - \tilde{c}_\alpha^B \delta_{ki}\delta_{nB} \end{bmatrix}$$

for $k, j = 1, .., d$ labeling dimensions, $\alpha = 0, ..., d$ labeling the extended coordinates, and $n = a, b, ...$ labeling the extended bodies. Similarly for the rotational axis restriction.

## E Benchmark systems

Below we detail a series of more challenging synthetic physical systems to benchmark our approach and the baselines. While the information of the Hamiltonian is withheld from our model, we detail the true Hamiltonians below that can be used to generate the data. Even though the equations of motion are very complex, the Hamiltonians in Cartesian coordinates are very simple, again demonstrating why this approach simplifies the learning problem.

### E.1 $N$-Pendulum

N point masses are connected in a chain with distance constraints $(\mathbf{0}, \mathbf{x}_1), (\mathbf{x}_1, \mathbf{x}_2), ..., (\mathbf{x}_{N-1}, \mathbf{x}_N)$ and the Hamiltonian is just the contributions from the kinetic energy and gravity in 2 dimensions:

$$H = \sum_n \mathbf{p}_n^\top \mathbf{p}_n / 2m_n + gm_n x_{n,2}. \tag{22}$$

## E.2 $N$-Coupled pendulums

In this 3 dimensional system, N point masses are suspended in parallel and springs connect the neighbors horizontally. The distance constraints are $(\mathbf{v}, \mathbf{x}_1), (2\mathbf{v}, \mathbf{x}_2), (3\mathbf{v}, \mathbf{x}_3), ..., (N\mathbf{v}, \mathbf{x}_N)$ where $\mathbf{v} = [1, 0, 0]^\top$ is a horizontal translation (of the origin). The Hamiltonian is:

$$H = \sum_{n=1}^{N} (\mathbf{p}_n^\top \mathbf{p}_n / 2m_n + gm_n x_{n,2}) + \sum_{n=1}^{N-1} \frac{1}{2} k (\|\mathbf{x}_n - \mathbf{x}_{n+1}\| - \ell_0)^2 \tag{23}$$

where $\ell_0 = \|\mathbf{v}\| = 1$.

## E.3 Magnetic pendulum

A magnet is suspended on a pendulum (in 3-dimensions) over a collection of magnets on the ground. The pendulum chaotically bounces between the magnets before finally settling on one of them.

Each of the magnets are modeled as dipoles with moments $\mathbf{m}_i \in \mathbb{R}^3$. The Hamiltonian for the system is

$$H(x, p) = \mathbf{p}^\top \mathbf{p} / 2m - \mathbf{m}_0(x)^\top \mathbf{B}(x) \tag{24}$$

where

$$\mathbf{B}(x) = \sum_i L(\mathbf{x} - \mathbf{r}_i)\mathbf{m}_i \text{ and } L(r) = \frac{\mu_0}{4\pi\|\mathbf{r}\|^5} (3\mathbf{r}\mathbf{r}^\top - \|\mathbf{r}\|^2 I), \tag{25}$$

$\mathbf{m}_0(x) = -q\frac{\mathbf{x}}{\|\mathbf{x}\|}$, $\mathbf{m}_i = q\hat{\mathbf{z}}$ for some magnet strength $q$. $\mathbf{r}_i$ are the spatial arrangements of the magnets placed on the plane. The constraints are just the distance constraint $(\mathbf{0}, \mathbf{x})$.

## E.4 Gyroscope

Consider a spinning top that contacts the ground at a single point. To simplify the learning problem, we choose the control points $x_i$ as unit vectors from the center of mass along the principle axes of the top. The Hamiltonian for the system is $H(x, p) = T + V = \mathrm{Tr}(PM^{-1}P^\top)/2 + mgX_{30}$. where

$$M^{-1} = \frac{1}{m} \begin{bmatrix} 1 & 1 & 1 & 1 \\ 1 & 1 + 1/\lambda_1 & 1 & 1 \\ 1 & 1 & 1 + 1/\lambda_2 & 1 \\ 1 & 1 & 1 & 1 + 1/\lambda_3 \end{bmatrix}. \tag{26}$$

Here we calculate the ground truth moments from the object mesh shown in Figure 4(d). In addition to the rigid body constraints, we simply need to add a universal joint connected to the origin.

## E.5 Rigid rotor

The Hamiltonian of this system in Cartesian coordinates is just the kinetic energy: $H(X, P) = \mathrm{Tr}(PM^{-1}P^\top)/2$. Like for the Gyroscope, we compute the ground truth moments from the object mesh shown in Figure 4(e).

# F Simplicity of Cartesian coordinates

## F.1 Gyroscope

The Hamiltonian of a gyroscope in Euler angles $(\phi, \theta, \psi)$ is given by

$$\mathcal{H} = \frac{1}{2} p^T M(\phi, \theta, \psi)^{-1} p + mg\ell \cos\theta \tag{27}$$

where the matrix $M$ can be derived by expanding out $\sum_i \mathcal{I}_i \omega_i^2$ using the angular velocity

$$\omega = [\dot{\phi}\sin\theta\sin\psi + \dot{\theta}\cos\psi, \dot{\phi}\sin\theta\cos\psi - \dot{\theta}\sin\psi, \dot{\phi}\cos\theta + \dot{\psi}]^\top$$

yielding the matrix

$$M = \begin{bmatrix} \sin^2\theta(\mathcal{I}_1\sin^2\psi + \mathcal{I}_2\cos^2\psi) + \cos^2\theta I_3 & (\mathcal{I}_1 - \mathcal{I}_2)\sin\theta\sin\psi\cos\psi & \mathcal{I}_3\cos\theta \\ (\mathcal{I}_1 - \mathcal{I}_2)\sin\theta\sin\psi\cos\psi & \mathcal{I}_1\cos^2\psi + \mathcal{I}_2\sin^2\psi & 0 \\ \mathcal{I}_3\cos\theta & 0 & \mathcal{I}_3 \end{bmatrix}. \quad (28)$$

Meanwhile, the Hamiltonian in Cartesian coordinates is given by the simpler form $H(x,p) = T + V = \mathrm{Tr}(PM^{-1}P^\top)/2 + mgX_{30}$, where

$$M^{-1} = \frac{1}{m}\begin{bmatrix} 1 & 1 & 1 & 1 \\ 1 & 1 + 1/\lambda_1 & 1 & 1 \\ 1 & 1 & 1 + 1/\lambda_2 & 1 \\ 1 & 1 & 1 & 1 + 1/\lambda_3 \end{bmatrix}. \quad (29)$$

## F.2   $N$-Pendulum

Suppose we have $N$ linked pendulums in two dimensions indexed from top to bottom with the top pendulum as pendulum $j$. Each pendulum $j$ has mass $m_j$ and is connected to pendulum $j-1$ by a rigid rod of length $l_j$. Let positive $y$ correspond to up and positive $x$ correspond to right. Then

$$x_j = \sum_{k=1}^{j} l_k \sin\theta_k = \sum_{k=1}^{N} \mathbb{1}_{[k\leq j]} l_k \sin\theta_k$$

$$y_j = -\sum_{k=1}^{j} l_k \cos\theta_k = -\sum_{k=1}^{N} \mathbb{1}_{[k\leq j]} l_k \cos\theta_k$$

$$\dot{x}_j = \sum_{k=1}^{j} l_k \dot{\theta}_k \cos\theta_k = \sum_{k=1}^{N} \mathbb{1}_{[k\leq j]} l_k \dot{\theta}_k \cos\theta_k$$

$$\dot{y}_j = \sum_{k=1}^{j} l_k \dot{\theta}_k \sin\theta_k = \sum_{k=1}^{N} \mathbb{1}_{[k\leq j]} l_k \dot{\theta}_k \sin\theta_k$$

$$\frac{\partial \dot{x}_j}{\partial \dot{\theta}_i} = \mathbb{1}_{[i\leq j]} l_i \cos\theta_i$$

$$\frac{\partial \dot{y}_j}{\partial \dot{\theta}_i} = \mathbb{1}_{[i\leq j]} l_i \sin\theta_i$$

Then given that $T = \sum_{j=1}^{N} \frac{1}{2} m_j (\dot{x}_i^2 + \dot{y}_i^2)$, we have that

$$
\begin{aligned}
p_i &= \frac{\partial T}{\partial \dot{\theta}_i} \\
&= \sum_{j=1}^{N} m_j \left( \dot{x}_j \frac{\partial \dot{x}_j}{\partial \dot{\theta}_i} + \dot{y}_j \frac{\partial \dot{y}_j}{\partial \dot{\theta}_i} \right) \\
&= \sum_{j=1}^{N} m_j \left( (\sum_{k=1}^{N} \mathbb{1}_{[k \leq j]} l_k \dot{\theta}_k \cos \theta_k)(\mathbb{1}_{[i \leq j]} l_i \cos \theta_i) + (\sum_{k=1}^{N} \mathbb{1}_{[k \leq j]} l_k \dot{\theta}_k \sin \theta_k)(\mathbb{1}_{[i \leq j]} l_i \sin \theta_i) \right) \\
&= \sum_{j=1}^{N} m_j \mathbb{1}_{[i \leq j]} \left( \sum_{k=1}^{N} \mathbb{1}_{[k \leq j]} l_k l_i (\cos \theta_k \cos \theta_i + \sin \theta_k \sin \theta_i) \dot{\theta}_k \right) \\
&= \sum_{j=1}^{N} \sum_{k=1}^{N} m_j \mathbb{1}_{[k \leq j]} \mathbb{1}_{[i \leq j]} l_i l_k \cos(\theta_i - \theta_k) \dot{\theta}_k \\
&= \sum_{k=1}^{N} \sum_{j=1}^{N} m_j \mathbb{1}_{[k \leq j]} \mathbb{1}_{[i \leq j]} l_i l_k \cos(\theta_i - \theta_k) \dot{\theta}_k \\
&= \sum_{k=1}^{N} \left( l_i l_k \cos(\theta_i - \theta_k) \sum_{j=1}^{N} m_j \mathbb{1}_{[k \leq j]} \mathbb{1}_{[i \leq j]} \right) \dot{\theta}_k \\
&= \sum_{k=1}^{N} \left( l_i l_k \cos(\theta_i - \theta_k) \sum_{j=\max(i,k)}^{N} m_j \right) \dot{\theta}_k
\end{aligned}
$$

where we made use of the fact that $\cos \theta_k \cos \theta_i + \sin \theta_k \sin \theta_i = \cos(\theta_i - \theta_k)$. To obtain the entries of the mass matrix, we match the above equation to for $p_i$ with the expected form

$$
p_i = \sum_{k=1}^{N} M_{ik} \dot{\theta}_k
$$

which gives

$$
M_{ik} = l_i l_k \cos(\theta_i - \theta_k) \sum_{j=\max(i,k)}^{N} m_j.
$$

**Equations of motion for the $N$-Pendulum.** For the $N = 2$ case, the equations of motion in generalized coordinates are

$$
\dot{q}_1 = \frac{\ell_2 p_1 - \ell_1 p_2 \cos(q_1 - q_2)}{\ell_1^2 \ell_2 (m_1 + m_2 \sin^2(q_1 - q_2))} \qquad \dot{q}_2 = \frac{-m_2 \ell_2 p_1 \cos(q_1 - q_2) + (m_1 + m_2) \ell_1 p_2}{m_2 \ell_1 \ell_2^2 (m_1 + m_2 \sin^2(q_1 - q_2))}
$$

$$
\dot{p}_1 = -(m_1 + m_2) g \ell_1 \sin \theta_1 - C_1 + C_2 \qquad \dot{p}_2 = -m_2 g \ell_2 \sin q_2 + C_1 - C_2
$$

where

$$
C_1 = \frac{p_1 p_2 \sin(q_1 - q_2)}{\ell_1 \ell_2 (m_1 + m_2 \sin^2(q_1 - q_2))} \qquad C_2 = \frac{m_2 \ell_2^2 p_1^2 + (m_1 + m_2) \ell_1^2 p_2^2 - 2 m_2 \ell_1 \ell_2 p_1 p_2 \cos(q_1 - q_2)}{2 \ell_1^2 \ell_2^2 (m_1 + m_2 \sin^2(q_1 - q_2))^2 / \sin(2(q_1 - q_2))}.
$$

For $N = 3$, *the equations of motion would stretch over a full page*.

On the other hand the equations of motion in Cartesian coordinates are described simply by

$$
\dot{x}_{i,1} = \frac{p_{i,1}}{m_i} \qquad \dot{x}_{i,2} = \frac{p_{i,2}}{m_i} \qquad \dot{p}_{i,1} = 0 \qquad \dot{p}_{i,2} = g m_i
$$

which maintain the same functional form irrespective of $N$.



[Supplementary Material 2 · perturbation.pdf]



2-Pendulum

Prediction | Data efficiency

$y_2$ axis: $-1$, $-2$

Rel. err. axis: $10^{-1}$, $10^{-5}$

Time (seconds): $0$, $10$

$N_{train}$: $10^2$, $10^4$

Geom. mean. of rel. err.: $10^{-1}$, $10^{-3}$, $10^{-5}$

Legend: GT, GT+$\varepsilon$, CHNN, CLNN, HNN, DeLaN, NeuralODE

[Supplementary Material 3]



Prediction over 100 timesteps

2D Systems

3D Systems

Geom. mean. of rel. err.

1-Pendulum  2-Pendulum  3-Pendulum  5-Pendulum

3-CoupledPendulum  Gyroscope  MagnetPendulum  Rotor

CHNN    CLNN    DeLaN    HNN    NeuralODE

[Supplementary Material 4 · energy-conservation.pdf]



1-Pendulum  2-Pendulum  3-CoupledPendulum  3-Pendulum

4-CoupledPendulum  5-Pendulum  Gyroscope  MagnetPendulum

Rel. err. in true energy

Time (seconds)

—— CHNN  —— CLNN  —— DeLaN  —— HNN  —— NeuralODE

[Supplementary Material 5]



**2-Pendulum**

Prediction | Data-efficiency

Legend: GT, GT+$\varepsilon$, CHNN, CLNN, HNN, DeLaN, NeuralODE

Axes: $y_2$, Rel. err., Time (seconds), $N_{train}$, Geom. mean. of rel. err.