[Reviews · NeurIPS 2020]

Review 1

Summary and Contributions: In this paper, the authors propose an approach to learning physical dynamical systems using NNs via optimization under constraints in Cartesian coordinates. They discuss the effectiveness of their approach compared with the existing Hamiltonian and Lagrangian NNs, which learn systems in generalized coordinates with implicit constraints.

Strengths: Their approach is based on the idea of enforcing conditions that have to be followed explicitly in the procedure where differential equations with Hamiltonian systems through transforming the objective to non-constrained one with Lagrange multipliers. This sounds reasonable and follows the well-established principles both in physics, ML and others

Weaknesses: The approach, which incorporate constraints in Cartesian coordinates through transforming the objective to non-constrained one with Lagrange multipliers, sounds somehow straightforward. Also, the procedure is restricted to physical systems, typically, in our 3d space (or 2d) because we need the calculation of the Hamiltonian systems although I admit this restricted focus is still an important problem.

Correctness: The claims and method in the paper seem to be correct. ----- Thank you for your rebuttal. I found misunderstanding in my first review.

Clarity: I think the paper does not include several necessary information. For example, I wonder what is a guideline to to design f_\theta with NNs given an objective physical system.

Relation to Prior Work: The paper describes the relation to existing works in the paper.

Reproducibility: No

Additional Feedback:


Review 2

Summary and Contributions: The authors show certain advantages of using explicit constraints in learning physical systems.

Strengths: The approach appears sound and has an impressive performance.

Weaknesses: There is a bigger picture question: In classical mechanics, we typically try to avoid the explicit constraints as they are numerically more brittle. Why is that not the case for the presented approach? Also, the competing approaches were evaluated on complex real world systems, e.g., DeLaN was evaluated on a 7 DoF robot for control. Why should your results be believable?

Correctness: The paper appears correct.

Clarity: Only experts on the topic can follow the paper. More Background is needed.

Relation to Prior Work: Yes

Reproducibility: Yes

Additional Feedback:


Review 3

Summary and Contributions: The authors propose a parametrization to learn Hamiltonian or Lagrangian dynamics using neural networks. Their idea is to modify the variational problem of Hamilton's principle so that the expression in the Cartesian coordinates satisfy the constraints required from the problem. Finally, the Hamiltonian (and Lagrangian) mechanics are modified with the projection operator so that the resulting trajectories satisfy the constraints. They show the effectiveness of the proposed formulation with variety of numerical examples.

Strengths: The idea is simple yet reasonable, and the experimental materials are quite convincing to understand the validity of the proposed parameterization. I believe this work is very important in the area of learning physical systems using neural networks.

Weaknesses: Overall, I did not find big flaws when reading.

Correctness: The method and the empirical methodology indeed seem valid and correct.

Clarity: The paper is very clear. I really enjoyed reading the paper.

Relation to Prior Work: I think the related work is sufficiently mentioned.

Reproducibility: Yes

Additional Feedback: Line 211: What V(X) denotes? I could not find where it was defined. Sorry if I just missed it somewhere. If I should say something more here, I would say that negative examples, if any, could make the discussion more complete. I understand that the proposed parameterization, seemingly just an incremental modification, is advantageous in many problems of physics learning as the Cartesian coordinate is an orthodox representation. Meanwhile, could you elaborate on cases where the proposed parameterization is not really advantageous and constitute such examples numerically? ----- Thank you for the rebuttal. As my evaluation is originally positive and I posed no serious concerns, I leave my score just unchanged.


Review 4

Summary and Contributions: In this paper, the Authors explicit use of the cartesian coordinates to represent the state of dynamical systems. They use this Cartesian representation to then derived either the Hamiltonian or the Langrangian related to the system dynamics. They derive the mathematics for both cases and shows that they might handle many complex systems not only in 3d but any dimensional space (which is required for problem going beyond mechanics).

Strengths: The proposed approach is very sound. The mathematics and their derivations are classic and very standard (maybe, they should appear in the appendix instead of section 5).

Weaknesses: It would have been nice to have a complete picture of the proposed approach to really describe the whole contribution. For instance, the learning problem is detailed at various places in the paper, making it hard to get the whole picture. I would rather suggest the Authors reorganizing their paper to get something smoother in terms of reading. I might appear that the matrix in Eq (5) and (6) might be singular. How do the Authors handle it in their numerical implementation?

Correctness: The experimental validation seems to contain sufficient comparison to show the effectiveness of the approach. They illustrate their approach to various dynamical systems. It would have been nice to really show the effectiveness on a robotic system like a 6d arm which may evolve in 3d with hard joint constraints.

Clarity: The paper is clear and sounds. I do think that the paper organization can be largely improved by moving some results (like the energy one) to the main corpus and reducing the size of the introduction (which seems a bit longer with too many details that may be postponed to the appendix). It seems that line 188, the notation differs from Eq. (7). The summary line 206 is not clear and should be revised.

Relation to Prior Work: The difference between the existing state of the art approaches is clearly stated.

Reproducibility: Yes

Additional Feedback:

[Author Response · NeurIPS 2020]

We thank the reviewers for their supportive feedback. We are glad that the reviewers found our method's performance to be "impressive" (R2), and the approach to be "very sound" (R4) and "very important in the area of learning physical systems" (R3). Our main contributions: (1) we show both analytically and empirically that using generalized coordinates to enforce constraints complicates the learning problem, and hinders generalization, (2) we propose a method that circumvents these problems by embedding into Cartesian coordinates and enforcing constraints explicitly in the learned Hamiltonian or Lagrangian dynamics, and (3) we empirically show that our approach is up to *100 times more data-efficient and up to 260 times more accurate* than previous methods, and outperforms previous methods on all systems. Our models' large gains in performance highlights the practical importance of representing the systems' dynamics in the simplest functional form possible, which has not been considered in previous works.

[R1] **"The approach, which uses regularization in Cartesian coordinates, sounds somehow straightforward."** The summary of our approach as "regularization in Cartesian coordinates" is incorrect. Our method is not regularization and we never even mention "regularization" in the paper. Unlike a regularized model, the constrained equations of motion we derive in section 5 *exactly* preserve the constraints, regardless of the learned Hamiltonian $\mathcal{H}$. In Appendix B3 we show that numerical integration of these equations preserves the constraints up to numerical tolerances. We also emphasize that in the context of outstanding results, being "straightforward" is a strength, not a weakness. Our method results in $\sim 100$ times more accurate and data-efficient models. Additionally, we believe working out how to embed the dynamics of complex 3D systems entirely into the evolution of Euclidean coordinates under constraints (without using angles, quaternions, etc) is also a substantial contribution.

[R1] **"The procedure is restricted to physical systems."** Indeed, our approach is restricted to systems in physical space, as we mentioned in line 285. Such systems are prevalent in engineering and robotics since our world is one such system. Our goal is to demonstrate how to learn the dynamics of such common physical systems more effectively.

[R1] **"the paper does not include several necessary information [for reproducibility]"** We detail the networks architectures in appendix D2, which are 3-layer MLPs. While we do reference this in the main text, some of the relevant details are distributed across the text, and so we will bring these together when summarizing the approach. Regarding reproducibility, our paper checks off every relevant item from the NeurIPS reproducibility checklist: we run multiple trials with error bars, release the source code, and include all hyperparameters, tuning procedures, and detailed derivations in the appendix.

[R2] **"In classical mechanics, we typically try to avoid the explicit constraints as they are numerically more brittle."** While evolving dynamics that explicitly enforce constraints numerically can lead to small accumulation of constraint drift, we quantify this drift in appendix B3, and we feel that the superior modeling ability more than makes up for this slight downside. Furthermore, the generalized coordinate approach of enforcing constraints comes with its own set of numerical instabilities, such as coordinate singularities and gimbal lock, that do not affect our proposed explicitly constrained models.

[R2,R4] **Testing on robotic systems and real world systems.** We agree this is an interesting direction for future work. We note that overall we perform a relatively exhaustive empirical evaluation, including the introduction of complex physical systems that challenge current approaches to learning Hamiltonians and Lagrangians. Developing a test setup for a robotic system is a considerable undertaking, and among the related methods of NeuralODE, HNN, DeLaN, LNN, SymODEN, SRNN and LieConv, only DeLaN was tested on real world mechanical data. In this paper we focus on simulated data which may also make it easier for others to build on our work and reproduce our results.

[R3] **"cases where the proposed parameterization is not really advantageous."** The approach is limited to systems in physical space which we briefly discussed in the conclusion. Another limitation is that the form of the constraints $\{\Phi(x)_i\}$ must be known ahead of time, such as the joint connectivity. We view the possibility of learning these constraints from data as a promising direction for future work. We will expand on these discussions.

[R3] **"What $V(X)$ denotes?"** $V(X)$ is the potential energy of the Hamiltonian or Lagrangian. We will clarify this.

[R4] **"the matrix in Eq (5) and (6) might be singular."** Instead of numerically inverting the matrix $M$, we use the closed form expression for the inverse given on line 195, and values are bounded away from $0$ because $m$ and $\lambda$ are parametrized using a $\mathrm{Softplus}$, so $M$ is never singular. Note that since the constraints $\{\Phi(x)_i = 0\}_{i=1}^{C}$ are assumed to be independent, $D\Phi$ has linearly independent columns, and therefore the matrix $\left[D\Phi M^{-1} D\Phi^{\top}\right]$ is invertible. The same holds for $\left[D\Psi J D\Psi^{\top}\right]$ in CHNNs.

**Clarity.** We appreciate that reviewers found the exposition largely clear. We will include some additional background, and move some material as suggested. We will also revise the summary after line 206, and fix the typo in line 188.

[Meta-Review · NeurIPS 2020]

The paper proposes to learn physical systems by embedding the system into Cartesian coordinates with explicit constraints. The reviewers agree the paper is novel but also suggest adding real-world data experiments.